# NONPARAMETRIC EXPERT DAG LEARNING WITH ACCURATE EDGE STRENGTHS AND REALISTIC KNOWLEDGE INCORPORATION

## ABSTRACT

Directed Acyclic Graphs (DAGs) are crucial for modeling causal structures and complex dependencies in domains such as biology, healthcare, and finance. Effective structure learning must not only align with domain expert knowledge but also produce interpretable model decisions. Though continuous structure learning methods like NOTEARS are gaining popularity, an underexplored feature is their ability to open up the black box of decisions made by traditional combinatorial search by quantifying edge strengths in weighted adjacency matrices. Yet challenges persist in systematically integrating expert knowledge and ensuring learned weights accurately reflect true edge relationships. We present Non-parametric Expert DAG (NEDAG), a novel method that formulates accurate weight matrices using Gaussian Processes (GPs) and incorporates realistic domain knowledge into the continuous structure learning framework. Experiments on both synthetic and real-world datasets demonstrate that NEDAG not only surpasses existing methods in structure accuracy but also produces more accurate edge strengths. NEDAG thus provides a robust and interpretable solution for structure discovery in real-world applications.

## 1 INTRODUCTION

Directed Acyclic Graphs (DAGs) are critical tools for modeling complex dependencies and causal structures in various domains such as healthcare (Lucas et al.), biology (Sachs et al.), and finance (Sanford & Moosa).

For DAGs to be useful to domain experts, several key features are essential. First, **expert knowledge should be incorporated into the model**. Since expert knowledge captures the understanding of the field, DAG learning should align with, rather than overwrite, prior knowledge. Moreover, prior knowledge alleviate the issue that from data, the DAG structure is often identifiable only up to its Markov equivalence class (Ghoshal & Honorio). Additionally, to solve the NP hard problem, (Chickering et al.), approximate search methods are frequently used, which often suffer from non-convexity (Chickering; Zheng et al., a). Leveraging expert knowledge can narrow down the search space and improve model accuracy by guiding the process closer to the global optimum, especially in data-scarce scenarios.

Second, **reliance on expert-specified parameters and distributional assumptions should be minimized.** Solicitation of distribution or functional form assumptions from experts, as often requires by parametric methods, necessitates significant expertise and risks misspecification (Zheng et al., b).

Third, **the structure learning process should be interpretable rather than opaque**. It is important that the reasons for including or excluding certain edges are transparent, allowing domain experts to understand the rationale behind the model's decisions. This often requires that the parameters which the models learns and makes decision based on are consistent with criteria meaningful for humans.

Traditional score-based, combinatorial structure learning methods, which search through the DAG space to minimize an objective function, have incorporated the first two of these principles. Expert knowledge in the form of required or forbidden edges has been used to constrain the search space (Constantinou et al.; Cooper & Herskovits; de Campos & Castellano; Ma et al.). Nonparametric

methods like Gaussian Processes (GPs), which do not require predefined parameters, have also been applied (Weng & Doshi-Velez; Atienza et al., b; Boukabour & Masmoudi). However, the learning process in these methods often remains opaque. In combinatorial search, local decisions about adding, removing, or reversing edges are made without clear visibility into their global impact, only revealed once the global objective is minimized (Chickering; Heckerman et al.).

A different formulation of DAG learning holds promise in opening up the black-box. Recent work by Zheng et al. (a) and others (Bello et al.; Yu et al., b; Massidda et al.; Ng et al.) introduced continuous structure learning approaches, where DAGs are represented through weighted adjacency matrices.

While still underexplored, continuous structure learning shows great potential to fulfill the three key principles for applied DAG learning. The weighted adjacency matrix formulation presents a unique opportunity to increase transparency by directly revealing edge strengths as matrix weights, effectively opening up the black box of model decisions on learned structures. Furthermore, the matrix-based representation naturally facilitates the integration of domain knowledge, which remains underutilized, particularly when combined with nonparametric methods.

In this work, we address each of the three principles for applied DAG learning within the continuous learning framework. First, we systematically integrate multiple, realistic forms of knowledge: Required Edges (REQ-EDG), Initial Graphs (INI-GRA), and (partial) Topological Orderings (TOP-ORD). Second, we integrate nonparametric method Gaussian Processes (GPs) in continuous learning, which reduces reliance on expert-specified parameters. Third, we present the first nonlinear weighted adjacency matrix formulation that accurately represents edge strength, improving the accuracy of parameter and structure learning, while making the model's rationale for edge selection more transparent to experts.

By integrating expert knowledge with accurate Gaussian Process weights, we develop the Nonparametric Expert DAG-GP (NEDAG-GP). Our contributions are twofold:

1. We enhance DAG learning by integrating diverse and realistic expert knowledge into the continuous learning framework (§4.2). By employing a fine-grained knowledge setup across various conditions, we reveal patterns that would otherwise be obscured by population averages. We demonstrate that incorporating knowledge significantly improves learning accuracy in both synthetic and real-world scenarios, such as Gene Regulatory Networks (§6).

2. We present NEDAG-GP, the first nonparametric method in continuous structure learning that represents edge strength more accurately than existing nonlinear methods. Proven theoretically (§4.1) and validated empirically (§6), it significantly boosts both DAG learning accuracy and interpretability.

## 2 RELATED WORK

### 2.1 CONTINUOUS LEARNING

In continuous learning, DAGs are represented through weighted adjacency matrices. This approach has been developed to handle both linear (Massidda et al.; Ng et al.) and nonlinear models (Zheng et al., b; Bello et al.; Yu et al., b; Lachapelle et al.). However, two key challenges remain:

**1. Inaccurate Weights in Nonlinear Models**

Defining a weighted adjacency matrix that accurately reflects edge strengths in nonlinear models, such as multilayer perceptrons (MLPs), is challenging, especially when the nonlinear functions lack a closed-form expression for edge strength (see definition of edge strength in 3.2). Most existing methods concede to using weighted adjacency matrices that only guarantee binary performance; that is, zero and nonzero entries represent the absence and presence of edges, respectively, but the magnitude of the weights does not correlate with edge strength. For instance, NOTEARS-MLP and DAGMA use the first-layer weights of an MLP to represent edge strength (Zheng et al., b; Bello et al.). This effectively binary matrix can result in arbitrarily large discrepancies between the learned weights and the true edge strengths (see A for proof), as demonstrated by Waxman et al., who proposed an approximation to partially mitigate this issue. This inaccuracy introduces three key issues: (i) inaccurate structure selection based on edge weights; (ii) lack of interpretability of

weights as edge strengths and of model decisions; and (iii) suboptimal optimization due to a weak correlation between weights and edge strengths.

**2. Lack of Nonparametric Methods in Continuous Learning** Another limitation in continuous DAG learning is the reliance on parametric methods. Parametric methods, for instance MLP (Zheng et al., b) or GNN (Yu et al., a), often require expert input to specify model architectures, functional and distributional assumptions, as well as tunable hyperparameters, placing an undue burden on users and leading to the risk of model misspecification. For instance, experts may oversimplify complex relationships by assuming linear DAGs, or when they opt for more flexible nonlinear models, they must choose from a wide range of potential specifications. Moreover, parametric models may suffer performance degradation when their assumptions are violated.

In this work, we address both challenges by introducing Nonparametric Expert DAG-Gaussian Processes (NEDAG-GP), the first nonparametric method with accurate weight formulation within the continuous learning framework (see 4.1). Unlike parametric methods like MLP, our GP formulation does not rely on expert specification of parameters and offers improved accuracy in representing true edge strengths, outperforming existing methods on both model interpretation and structure learning (see 6).

## 2.2 Incorporating Prior Knowledge in Structure Learning

DAGs are widely used to represent causal structures and dependencies across domains, and incorporating prior knowledge can significantly enhance their construction. For example, in Gene Regulatory Networks (GRNs), domain experts often have insights into gene interactions that can guide the structure learning process (Sachs et al.).

In combinatorial structure learning, prior knowledge has been incorporated through several mechanisms. The presence or absence of specific edges has been encoded as prior probabilities (Castelo & Siebes), embedded as rewards or penalties in the objective function (Heckerman et al.), or enforced as hard constraints that limit the search space (de Campos & Castellano). Additionally, topological orderings—whether full or partial—have been used to impose further constraints (Cano et al.; Cooper & Herskovits; Ma et al.).

In continuous structure learning, however, the integration of prior knowledge has seen limited exploration. Sun et al. enforce required and forbidden edges as hard constraints on nonzero and zero edge weights, respectively, in Dynamic Bayesian Networks. Similarly, Chowdhury et al. apply constraints sequentially, enforcing required or forbidden edges after model errors are identified. Focusing on the same knowledge types, Hasan & Gani encode required edges as 1 and forbidden edges as 0 in a reinforcement learning framework.

Existing knowledge incorporation in continuous structure learning is limited in the types of knowledge it can handle, the granularity it supports, and its applicability to real-world scenarios. For instance, prior knowledge isn't simply about specifying required or forbidden edges—edges can exist with varying degrees of confidence. Moreover, enforcing constraints on required versus forbidden edges may not be as straightforward as suggested by Chowdhury et al.; the effectiveness of these constraints likely depends on factors such as network sparsity and sample size. In applied domains, particularly those involving sparse graphs, experts often lack complete information about forbidden edges, making it unrealistic to demand such specifications. However, negative knowledge—regarding forbidden edges—can still exist in alternative forms, such as topological orderings that implicitly prohibit edges from lower-tier nodes to higher-tier nodes. This suggests the need for more nuanced approaches to soliciting and incorporating expert knowledge.

In this work, we systematically integrate more realistic prior knowledge into continuous structure learning (see §4.2), study their fine-grained effects under various conditions, and demonstrate their efficacy on synthetic datasets and a real-world GRN inference task (see §6).

## 2.3 Nonparametric Methods in DAG Learning

Nonparametric methods offer greater flexibility and are better suited for capturing complex, unknown relationships without the strict assumptions of parametric models. Approaches such as Gaussian Processes (GPs), Kernel Density Estimation, and the Nadaraya-Watson estimator have

been employed to augment oversimplified linear DAG models in combinatorial structure learning (Weng & Doshi-Velez; Atienza et al., a; Boukabour & Masmoudi).

Beyond the general advantages of nonparametric methods, additive Gaussian Processes (GPs) with an RBF kernel possess unique features that make them particularly suitable for formulating interpretable and accurate weights in continuous structure learning, especially when combined with expert knowledge.

The additive and smooth nature of GPs with RBF kernels enables the isolation of local influences (Friedman & Nachman), allowing individual edge strengths to be derived in closed form. Such GPs learn only two interpretable hyperparameters: *amplitude* (i.e., the significance of the dependence) and *length scale* (i.e., the distance over which dependence diminishes) (Luger et al.), both of which intuitively contribute to our derived edge weight formulation. These characteristics of additive GP with RBF kernels ensure that the derived weights accurately reflect edge strengths (see §4.1).

Furthermore, the probabilistic nature of GPs allows for seamless incorporation of expert knowledge as priors on these hyperparameters (Weng & Doshi-Velez).

Together, these features make additive GPs with RBF kernels particularly well-suited for continuous DAG learning, as they minimize the need for parametric assumptions from experts, provide accurate and interpretable weights, and facilitate the integration of domain knowledge.

## 3 BACKGROUND

### 3.1 STRUCTURE LEARNING

DAG structure learning aims to uncover the underlying graphical model from observed data. Let $X \in \mathbb{R}^{n \times d}$ represent $n$ i.i.d. observations of a random vector $X = (X_1, \ldots, X_d)$.

A nonparametric structural equation model (SEM) is defined by:

$$X_j = f_j(X, Z_j), \quad j \in [d], \tag{1}$$

where each $f_j : \mathbb{R}^{d+1} \to \mathbb{R}$ is a nonparametric function, and $Z_j$ are independent exogenous variables representing noise. Each $f_j$ depends only on a subset of $X$ (the parents of $X_j$) and $Z_j$, inducing a graphical structure $G(f)$, which we assume is a DAG. Our goal is to learn this graph $G(f)$ from data.

In score-based learning, a score function $L(f; X)$ evaluates the quality of a candidate SEM, which is the sum of loss-least squares or negative log-likelihood-and often regularized with penalties like BIC or $\ell_1$-norms. The structure learning problem then becomes:

$$\min_{f \in \mathcal{F}} L(f; X) \quad \text{subject to } G(f) \in \mathcal{D}, \tag{2}$$

where $\mathcal{D}$ is the space of DAGs on $d$ nodes, and $\mathcal{F}$ is a function space.

### 3.2 CONTINUOUS STRUCTURE LEARNING

Unlike the traditional approach that searches through the discrete space of DAGs $\mathcal{D}$, continuous structure learning operates in the continuous space of weighted adjacency matrices $W \in \mathbb{R}^{d \times d}$, where a directed edge $X_k \to X_j$ exists if and only if $w_{kj} \neq 0$.

A function $h(W)$ is introduced to enforce the DAG constraint, ensuring that $W$ represents a valid DAG. Yu et al. (a) proposed a polynomial constraint:

$$h_{\text{poly}}(W) = \text{Tr}\left( \left( I + \frac{1}{d} W \circ W \right)^d \right) - d, \tag{3}$$

where $\circ$ denotes the Hadamard product and $I$ is the identity matrix. This formulation prevents closed walks, a defining property of DAGs.

The problem is then formulated as:

$$\min_{f \in H^1(\mathbb{R}^d)} L(f) \quad \text{subject to } h(W(f)) = 0, \tag{4}$$

where $W$ represents the learned adjacency matrix.

At a minimum, the weight matrix $W$ should correctly represent the graph structure: zero weights indicate no edge, and nonzero weights imply the presence of an edge. However, in practice, weights rarely reach exact zeros, necessitating an additional step to retain only the significant weights, which is often thresholding in existing works. To retain the correct edges, the consistency between edge strengths and learned weights is required. Thus, ideally, $W$ should not only indicate the presence of edges but also accurately reflect their true strengths, defined as the $L_2$-norm of the partial derivative of $f_j$ with respect to $x_k$ Rosasco et al.:

$$[W(f)]_{kj} := \left\| \frac{\partial f_j}{\partial \mathbf{x}_k} \right\|_{L_2}. \tag{5}$$

In the following section, we review the fundamentals of GPs before presenting our method for defining such weights that accurately reflect true edge strengths.

### 3.3 ADDITIVE GAUSSIAN PROCESSES

An additive Gaussian Process (GP) models the function $f(x)$ as a sum of independent GPs, each corresponding to different input dimensions. For a single GP that encodes a single edge, it is defined by a mean function $m(x)$ and a covariance function $k(x, x')$, and can be expressed as $f(x) \sim \mathcal{GP}(m(x), k(x, x'))$.

In practice, the mean function is often assumed to be zero, centering the GP around zero. When the covariance function $k(x, x')$ is evaluated for specific input points, the function values follow a multivariate normal distribution: $\mathbf{f} \sim \mathcal{N}(\mathbf{0}, \mathbf{K})$, where $\mathbf{K}$ is the covariance matrix with $[\mathbf{K}]_{ij} = k(x_i, x_j)$.

A commonly used smooth covariance function is the Radial Basis Function (RBF) kernel, defined as:

$$k_{\text{RBF}}(x, x') = \sigma_f^2 \exp\left( -\frac{||x - x'||^2}{2\ell^2} \right), \tag{6}$$

where $\sigma_f^2$ controls the amplitude, and $\ell$ is the length scale, determining how quickly correlations decay with distance.

Additive GPs with RBF kernels allow for closed-form, interpretable edge strengths in DAG structure learning, as derived in §4.1.

## 4 METHOD

Our model is two-part, focusing on formulating interpretable GP weights, and incorporating knowledge, respectively.

### 4.1 FORMULATING GP WEIGHTS AS EDGE STRENGTHS

Current weight formulations in continuous structure learning often fail to ensure that non-zero weights correspond to accurate edge strengths. To address this, we develop a precise weight formulation for Gaussian Processes (GPs). In parametric methods, edge strength is typically defined as the $L_2$-norm of partial derivatives. However, for GPs—which model a distribution over functions—the $L_2$-norm can vary across different function realizations.

To extend this to GPs, we propose using the expected $L_2$-norm of the partial derivative, ensuring that the weighted adjacency matrix reflects the average influence of $x_k$ on $f_j$ across all GP realizations. This approach adapts the consensus definition of edge strength to the GP context, addressing the inherent variability in nonparametric models.

Since $x_k$ is continuous, the $L_2$-norm is defined as the integral of the squared function over the domain:

$$\mathbb{E}\left[\left\|\frac{\partial f_j}{\partial x_k}\right\|_{L_2}^2\right] = \int_{\mathcal{X}} \mathbb{E}\left[\left(\frac{\partial f_j(x)}{\partial x_k}\right)^2\right] dx \tag{7}$$

Here, $\mathbb{E}\left[\left(\frac{\partial f_j(x)}{\partial x_k}\right)^2\right]$ represents the variance of the partial derivative $\frac{\partial f_j(x)}{\partial x_k}$, which for an additive GP corresponds to the variance of the derivative with respect to a single parent $x_k$. This variance is computed from the GP covariance function $K(x, x')$. For the Radial Basis Function (RBF) kernel, the covariance of the partial derivative at $x = x'$ is:

$$\mathbb{E}\left[\left(\frac{\partial f_j(x)}{\partial x_k}\right)^2\right] = K_k(x, x') = \left.\frac{\partial^2 K(x, x')}{\partial x_k \partial x'_k}\right|_{x=x'} = \frac{\sigma^2}{\ell^2} \tag{8}$$

Substituting this into the integral for the expected $L_2$-norm gives $\frac{\sigma^2}{\ell^2} \text{Vol}(\mathcal{X})$. To match the interpretation in linear models, where the $L_2$-norm of the partial derivative is proportional to $\beta\sqrt{\text{Vol}(\mathcal{X})}$, we normalize by the square root of the domain volume $\sqrt{\text{Vol}(\mathcal{X})}$. This normalization ensures that GP-based weights are interpretable and comparable across different datasets, providing a consistent measure of edge strength.

Thus, the final weight formulation becomes:

$$[W(f)]_{kj} := \frac{\mathbb{E}\left[\left\|\frac{\partial f_j}{\partial x_k}\right\|_{L_2}\right]}{\sqrt{\text{Vol}(\mathcal{X})}} = \frac{\sigma}{\ell} \tag{9}$$

This formulation shows how the decomposability of additive GPs and the smoothness of RBF kernels allow for exact, closed-form edge strength expressions. Notice how, thanks to GP's interpretable hyperparameters, the derivation result shows that each edge strength can be represented through the learned hyperparameters in an intuitive way, with $\sigma$ (amplitude) and $\ell$ (length scale) reflecting the magnitude and range of dependencies, respectively.

## 4.2 KNOWLEDGE TYPES AND REPRESENTATION

| Knowledge | Definition | Examples in GRN | Constraint |
|---|---|---|---|
| **Required Edges (REQ-EDG)** | High confidence in the existence of specific edges. | Gene interactions experimentally validated with high confidence (consensus network). | Enforce $W_{ij} \geq \epsilon$, where $W_{ij}$ represents the weight of the required edge $i \to j$, and $\epsilon$ is a predefined threshold. |
| **Initial Graph (INI-GRA)** | Experts' best guess of the full graph structure, which can be updated. | Gene interactions that may be established or merely reported in some literature. | Initialize $W = W_{\text{init}}$, where $W_{\text{init}}$ is the expert-provided initial adjacency matrix. |
| **Topological Ordering (TOP-ORD)** | Grouping of variables into tiers, representing a (partial) topological order. | Genes grouped into master regulators, intermediates, and targets in GRNs. | Constrain $W_{ij} \leq \delta$ for edges $i \to j$ that violate the topological order, where $\delta$ is a small upper bound (e.g., close to 0). |

Table 1: Overview of the three knowledge types, their definitions, instances in GRNs, and the corresponding constraints in NEDAG.

Focusing on qualitative structural knowledge that is practical for experts to specify, we incorporate three types of knowledge: Required Edges (REQ-EDG), Initial Graph (INI-GRA), and (partial) Topological Orderings (TOP-ORD), adapted from Constantinou et al.. Our experiments utilize these

knowledge types based on information derived from synthetic datasets and the literature on Gene Regulatory Networks (GRNs) (Sachs et al.).

## 5 EXPERIMENTS

Our GP weight formulation and knowledge incorporation are general and can be integrated into any continuous learning framework. To maintain fairness in comparing our GP-based method with the MLP formulation, we primarily adopt the setup from NOTEARS (Zheng et al., b), including the learning algorithm, simulation setup (graph types, data models, sample sizes), and evaluation metrics (see B for details). While DAGMA offers speed advantages, we chose NOTEARS due to its flexible initialization conditions and comparable performance (Bello et al.).

Key adjustments to the NOTEARS framework include the introduction of an additive sine function, a nonlinear function with known closed-form edge strengths (see C for derivation), allowing us to evaluate the accuracy of the learned edge weights against a known ground truth.

Instead of relying on an arbitrary fixed threshold to remove edges, we select the top $e$ strongest edges, with $e$ determined by the specific dataset (refer to each dataset for the value of $e$). This selection method is further validated by the strong correlation between the learned weights of NEDAG-GP and the true edge strengths, as demonstrated in §6.

## 6 RESULTS

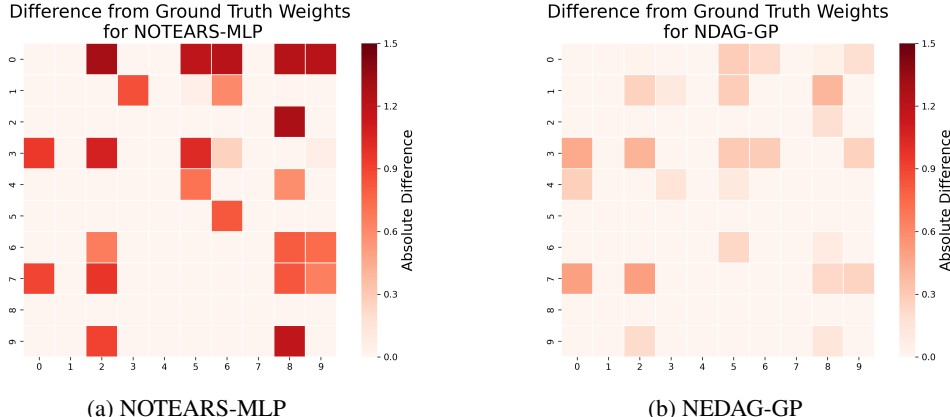

(a) NOTEARS-MLP                    (b) NEDAG-GP

Figure 1: Comparison of learned weight differences from true edge strengths for a random 10-node, 20-edge Erdős-Rényi graph. NEDAG-GP (right) shows significantly smaller deviations from the true values compared to NOTEARS-MLP (left), indicating superior edge strength capture by NEDAG-GP.

**NEDAG-GP Learns Edge Strengths More Accurately than NOTEARS-MLP**   Using simulated data from an additive sine function $f(x) = A\sin(Bx)$, where true edge strength is $\frac{AB}{\sqrt{2}}$ (see C for derivation), we evaluate the accuracy of NEDAG-GP's learned edge strengths compared to those of NOTEARS-MLP. NEDAG-GP's learned weights consistently approximate true edge strengths more closely, as shown in Fig 1 and Table 2.

Additionally, NEDAG-GP achieves a higher rank correlation with true edge strengths (Table 2), indicating a more accurate mapping between learned and true structures. This contrasts with NOTEARS-MLP, which shows weaker alignment with actual edge strengths, reaffirming the theoretical limitations of MLP-based weight formulations (See A).

The importance of absolute differences lies in the accurate recovery of true parameters, while the relative ranking of edges is crucial for guiding structure selection. This provides domain experts with clearer insights into which edges are most significant. NEDAG-GP's strong correlation between

learned and true edge strengths supports our decision to select the top strongest edges, rather than relying on arbitrary thresholds as in previous studies.

| Method | Difference | Ranking Correlation |
|---|---|---|
| NEDAG-GP | 7.531 ± 1.665 | 0.827 ± 0.093 |
| NOTEARS-MLP | 16.558 ± 2.441 | 0.740 ± 0.260 |

Table 2: NEDAG-GP outperforms NOTEARS-MLP in both weight accuracy and ranking correlation. Results reported as mean ± standard deviation.

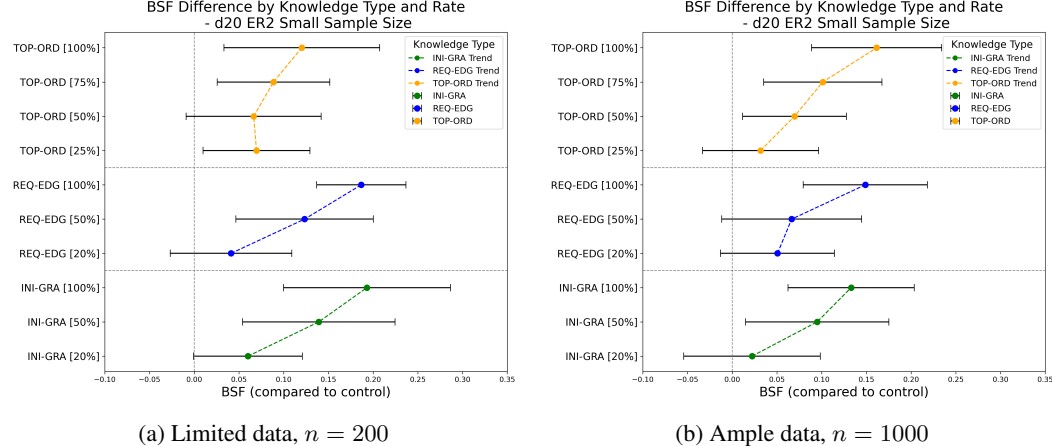

(a) Limited data, $n = 200$        (b) Ample data, $n = 1000$

Figure 2: Effect of prior knowledge on structure learning accuracy, measured by the Balanced Scoring Function (BSF, higher is better). The graphs compare the impact of three knowledge types (REQ-EDG, TOP-ORD, INI-GRA) across varying knowledge rates in both limited ($n = 200$) and ample ($n = 1000$) data scenarios. Higher knowledge rates consistently lead to improved performance, with the most effective knowledge type varying by dataset size. Error bars represent standard deviations.

**Higher Rates of All Knowledge Types Improve Learning in Small and Large Datasets** We first evaluate the performance of NEDAG-GP without knowledge in a wide range of graph types, data types and network sizes. We test NEDAG-GP on both additive GP and Additive Noise Model with MLP datasets across six graphs—Erdos-Rényi (ER) and scale-free (SF) with d = (5, 10, 20) nodes and 2d edges across two sample sizes ($n = 200$ and $n = 1000$). We keep only the top 2d significant edges. NEDAG-GP outperforms NOTEARS-MLP in the additive GP setting, while NOTEARS-MLP performs better on MLP data; as graph size increases, the learning challenge intensifies (see Appendix D).

Focusing on the most challenging scenario (ER with 20 nodes and 40 edges), we evaluate the effect of different types and levels of prior knowledge on NEDAG-GP performance across two sample sizes ($n = 200$ and $n = 1000$). Three knowledge types are tested: REQ-EDG (required edges), TOP-ORD (temporal order), and INI-GRA (initial graph), each at varying levels (0.2, 0.5, 1 for REQ-EDG/INI-GRA and 0.25, 0.5, 0.75, 1 for TOP-ORD). Higher rates are used for TOP-ORD as it is usually more accessible to experts (e.g., temporal sequences or upstream-downstream relationships), and even 100% knowledge only forbids edges violating the partial topological order, leaving flexibility within tiers.

Knowledge consistently improves structure learning across all conditions. However, our fine-grained analysis of the effects of different knowledge types across sample sizes reveals that the impact of knowledge varies with dataset size and type of knowledge. While smaller sample sizes are expected to benefit more from expert knowledge, the relationship is more nuanced, indicating additional influencing factors.

With limited data, REQ-EDG and INI-GRA yield the most significant gains, as these positive constraints guide the search toward better local minima. In larger datasets, negative constraints like

TOP-ORD have a greater impact. As the model has more data to uncover the true structure, forbidding edges may help prevent overfitting to spurious relationships.

Interestingly, REQ-EDG does not always outperform INI-GRA, despite REQ-EDG encompassing INI-GRA and maintaining the additional edges. This discrepancy might stem from suboptimal weighting of required edges, which could limit optimization in other parts of the graph.

Our results provide a more nuanced view compared to Chowdhury et al., who concluded that positive constraints are generally more useful for improving accuracy than negative constraints. This suggest the necessity of systematically studying the effects of knowledge types across diverse conditions to obtain a fuller understanding.

Overall, NEDAG-GP consistently benefits from the incorporation of expert knowledge, showing significant improvements across both small and large datasets. Higher rates of knowledge, whether REQ-EDG, TOP-ORD, or INI-GRA, progressively enhance performance. The effectiveness of each knowledge type varies with dataset size, demonstrating NEDAG-GP's robustness in addressing diverse structure learning challenges.

| Method | SHD $\downarrow$ | # Predicted Edges |
|---|---|---|
| **Baselines** | | |
| Empty Graph | 17 | 0 |
| NOTEARS-MLP | 16 | 13 |
| NoCurl + DAG-GNN | 16 | 18 |
| GOLEM | 14 | 11 |
| GraN-DAG | 13 | N/A |
| **NEDAG** | | |
| No Knowledge | 13 | 12 |
| + INI-GRA 50% | 14 | 12 |
| + REQ-EDG 50% | 11 | 13 |
| + TOP-ORD 3-tiers | 9 | 10 |
| + REQ-EDG 50% + TOP-ORD 3-tiers | 5 | 12 |

Table 3: SHD scores (lower is better) and predicted edges for different methods on the Sachs GRN. NEDAG-GP, enhanced with expert knowledge, achieves the best performance with a combination of REQ-EDG and TOP-ORD.

**Real-World Data: NEDAG-GP Excels in GRN Inference** We evaluate NEDAG-GP on the Sachs consensus Gene Regulatory Network (GRN), a real-world 11-node graph with 853 observational samples.

In real-world applications like this, expert knowledge is often readily available. Sachs et al. inferred a 17-edge model, 15 of which are supported by literature consensus. We incorporate these 15 established edges either as REQ-EDG or INI-GRA, while TOP-ORD is derived from categorical knowledge common in GRNs, dividing genes into three tiers: master regulators (which regulate others), intermediates, and targets (which can only be regulated). We select the top $e$ significant edges, where $e$ is optimized within a reasonable range around the 15 established edges (between 10 and 20). Consistent with previous studies (Zheng et al., a; Yu et al., a; Ng et al.; Lachapelle et al.), we report the SHD against the 17-edge graph from Sachs et al..

NEDAG-GP is compared against baselines including NOTEARS-MLP, NoCurl + DAG-GNN, GOLEM, and GraN-DAG. Without expert knowledge, NEDAG-GP achieves an SHD of 13, already outperforming most baselines. However, the real gains appear with expert knowledge incorporation: REQ-EDG reduces SHD to 11, and introducing a simple 3-tier TOP-ORD further reduces SHD to 9. Most notably, combining both REQ-EDG and TOP-ORD results in a substantial SHD reduction to 5—an over 60% improvement, marking the best performance across all methods.

This combination highlights that integrating both positive (REQ-EDG) and negative (TOP-ORD) constraints complements each other and significantly enhances model performance. Leveraging multiple knowledge types proves essential in tackling complex real-world networks like GRNs.

INI-GRA, however, shows no improvement (SHD = 14), likely due to suboptimal initial graph weights or its reduced effectiveness as a positive constraint in larger datasets, consistent with the synthetic data results.

Interestingly, TOP-ORD's ability to forbid edges results in better accuracy than positive constraints alone, again contradicting Chowdhury et al.. This might be because even a simple 3-tier TOP-ORD encodes information about a large number of forbidden edges, despite being easy for experts to specify. This finding underscores the importance of designing knowledge types that align with domain-specific realities, rather than focusing on mathematical convenience.

Overall, NEDAG-GP demonstrates strong capabilities in leveraging expert knowledge, producing interpretable edge weights that closely match true edge strengths. This leads to significant improvements in structure learning accuracy, with robust performance across both synthetic and real-world datasets.

## 7 DISCUSSION

This work demonstrates the potential of NEDAG-GP in leveraging expert knowledge to enhance both structure learning accuracy and interpretability, particularly in gene regulatory networks (GRNs) and other small-scale systems where domain expertise is critical and the interpretability of learned graphs is essential.

The two core goals of this paper—formulating accurate edge weights and systematically incorporating realistic expert knowledge—address important but underexplored principles in applied continuous DAG structure learning. These methods have been shown to effectively improve both learning accuracy and the interpretability of model decisions. Importantly, the principles and techniques developed here are generalizable and can be applied to any continuous learning framework or domain.

Although Gaussian Processes (GPs) come with computational and sample complexity challenges, they were chosen for this study due to their unique suitability for both tasks. Additive GPs with RBF kernels provide closed-form, interpretable edge strength representations due to their local decomposability, smoothness, and interpretable hyperparameters—such as amplitude and length scale. Furthermore, the probabilistic nature of GPs makes them well-suited for knowledge incorporation, such as through Bayesian priors on hyperparameters.

For future work on knowledge incorporation, our evaluation scheme sets the groundwork by offering a finer-grain, more realistic analysis of knowledge types. Our results emphasize the need to study the effects of different knowledge types under various conditions, as averaging across populations can obscure important patterns, similar to Simpson's Paradox. Additionally, our findings underscore the importance of designing knowledge incorporation methods that align with domain-specific needs, rather than being driven solely by mathematical convenience.

Taken together, these insights highlight the strong potential of GPs for future research, particularly in incorporating nuanced types of knowledge via Bayesian methods. While this study focused on more straightforward constraints on the weight matrix—which already showed strong performance, further investigation into GP-based models for domain-specific knowledge integration is a promising direction.

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

## A  NON-INTERPRETABILITY OF MLP WEIGHT FORMULATION IN CONTINUOUS LEARNING

A key motivation for NEDAG-GP is the non-interpretability of the weighted adjacency matrix $A_\theta$ in existing nonlinear methods. Specifically, methods like NOTEARS-MLP use weights derived from the first layer of an MLP, which may differ significantly from the true edge strengths, defined by the partial derivatives of child nodes with respect to parent nodes.

Let $f_j : \mathbb{R}^d \to \mathbb{R}$ be an MLP with weight matrices $A^{(1)}, A^{(2)}, \ldots, A^{(M)}$ and activation function $\sigma(\cdot)$. The MLP with $M - 1$ hidden layers can be expressed as:

$$f_j(\mathbf{x}) = A^{(M)} \sigma \left( A^{(M-1)} \sigma \left( \cdots \sigma \left( A^{(1)} \mathbf{x} \right) \right) \right).$$

It is known that:

$$\left\| \frac{\partial f_j}{\partial x_i} \right\|_{L2} = 0 \iff \left\| A^{(1)}_{i\cdot} \right\|_{L2} = 0,$$

but continuous structure learning relies on thresholding nonzero values, which can result in significant discrepancies. The following lemmas show that, under sigmoid activation, the two norms can differ arbitrarily.

**Lemma 1:** Let $\sigma(\cdot)$ be the sigmoid activation. For any $\delta, \epsilon > 0$, there exists an MLP $f_j$ with weight matrices $A^{(1)}, A^{(2)}, \ldots, A^{(M)}$ such that $\left\| A_{i\cdot}^{(1)} \right\|_{L2} < \epsilon$ but $\left\| \frac{\partial f_j}{\partial x_i} \right\|_{L2} > \delta$.

**Proof:** Consider a 1-hidden layer MLP $g(\mathbf{x}) = A^{(2)} \sigma\left(A^{(1)}\mathbf{x}\right)$. If $A^{(2)}$ is a $K \times H$ matrix, then:

$$[g(\mathbf{x})]_k = \sum_{h=1}^{H} A_{kh}^{(2)} \sigma\left(\sum_{j=1}^{d} A_{hj}^{(1)} x_j\right).$$

Let $z_h = \sigma\left(\sum_{j=1}^{d} A_{hj}^{(1)} x_j\right)$. Since $\sigma'(z_h) > 0$, applying the chain rule shows that $\frac{\partial z_h}{\partial x_i} > \epsilon$ for some $\epsilon > 0$. Therefore, by calculating the norm, $\left\| \frac{\partial f_j}{\partial x_i} \right\|_{L2}$ can be made larger than $\delta$, even when $\left\| A_{i\cdot}^{(1)} \right\|_{L2}$ is small.

**Lemma 2:** Let $\sigma(\cdot)$ be the sigmoid activation. For any $\delta, \epsilon > 0$, there exists an MLP $f_j$ with weight matrices $A^{(1)}, A^{(2)}, \ldots, A^{(M)}$ such that $\left\| A_{i\cdot}^{(1)} \right\|_{L2} > \epsilon$ but $\left\| \frac{\partial f_j}{\partial x_i} \right\|_{L2} < \delta$.

**Proof:** Similarly, for a 1-hidden layer MLP $g(\mathbf{x}) = A^{(2)} \sigma\left(A^{(1)}\mathbf{x}\right)$, if the entries of $A^{(1)}$ are large, then $z_h = \sum_{j=1}^{d} A_{hj}^{(1)} x_j$ becomes large. As $|z_h| \to \infty$, $\sigma'(z_h) \to 0$, so the gradients $\frac{\partial g(\mathbf{x})}{\partial x_i}$ become small. Thus, despite $\left\| A_{i\cdot}^{(1)} \right\|_{L2} > \epsilon$, we can ensure that $\left\| \frac{\partial f_j}{\partial x_i} \right\|_{L2} < \delta$.

# B  EXPERIMENTAL SETUP

## B.1  CONTINUOUS LEARNING FRAMEWORK

We adopt the NOTEARS framework of Zheng et al. (b), solving the problem as an unconstrained minimization after Lagrangian augmentation:

$$\min_{\theta} F(\theta) + \lambda \|\theta\|_1, \quad F(\theta) = L(\theta) + \frac{\rho}{2}|h(W(\theta))|^2 + \alpha h(W(\theta)),$$

where $\rho$ is a penalty parameter, $\alpha$ is a dual variable, and $h$ is the polynomial DAG constraint as defined earlier in 3.2. We solve this using the L-BFGS algorithm with $\ell_1$-regularization. The primary differences between our approach and NOTEARS-MLP lie in the model and weight matrix formulation.

## B.2  MODELS

**Nonparametric DAG-Gaussian Process (NEDAG-GP)**  We use a GP with an RBF kernel, with amplitude $\sigma$ and length scale $\ell$ initialized to 0.01 and 1, respectively. The weight is defined as $\sigma/\ell$, aligning with our interpretability objective. The loss function used is the negative log-likelihood.

NEDAG incorporates expert knowledge through constraints on required edges (REQ-EDGE), initial graphs (INI-GRA), and topological ordering (TOP-ORD). As outlined in Section 4.2, REQ-EDGE sets a lower bound of 1 for required edges, INI-GRA assigns a weight of 1 to pre-existing edges, and edges violating TOP-ORD are constrained to weights below 0.01.

## B.3  SIMULATIONS

We adopt the nonlinear setup from Zheng et al. (b), generating ground truth DAGs from Erdos-Rényi (ER) and scale-free (SF) random graph models. ER2 refers to an ER graph with $2d$ edges, similarly for SF. For each DAG, we simulate a structural equation model (SEM):

$$X_j = f_j(X_{\mathrm{pa}(j)}) + Z_j, \quad Z_j \sim \mathcal{N}(0, 1),$$

where $f_j$ varies across synthetic datasets, as described below.

**Synthetic Dataset with Ground Truth Edge Strengths**   We simulate an additive sine function:

$$f(X_i) = A \sin(B X_i),$$

where the true edge strength is $\frac{AB}{\sqrt{2}}$ (see C for derivation), allowing direct comparison of the weights learned by NEDAG-GP and NOTEARS-MLP with the known, closed-form edge strengths of the nonlinear sine function.

**Adversarial Dataset to Test Knowledge Incorporation**   We evaluate NEDAG-GP on two models: (1) additive models with GPs and (2) additive noise models (ANMs) with MLPs. NEDAG-GP performs well on additive GP data, but the MLP dataset is more challenging. We focus on MLP to assess how varying types and rates of prior knowledge in NEDAG-GP affects learning.

### B.4   BASELINES

For weight interpretability, we compare our GP-based formulation with the MLP formulation, which is the standard approach in nonlinear continuous structure learning, used in methods such as NOTEARS and DAGMA (Zheng et al., b; Bello et al.). While DAGMA improves optimization speed over NOTEARS-MLP, it requires the initial matrix to start within a specific feasible space (such as a zero matrix as a sufficient condition). This restriction makes DAGMA incompatible with our GP-based weight formulation, which depends on non-degenerate hyperparameters. As a result, we focus on comparing against NOTEARS-MLP, which, like DAGMA, uses an MLP formulation but allows for more flexible initialization conditions, making it a more suitable baseline for our approach.

For the evaluation of prior knowledge in synthetic datasets, we primarily compare NEDAG-GP with its expert knowledge-enhanced counterpart, NEDAG-GP. In the real-world dataset, NEDAG-GP is compared against multiple baselines, including NOTEARS-MLP, NoCurl + DAG-GNN (Yu et al., b), GOLEM (Ng et al.), and GraN-DAG (Lachapelle et al.).

### B.5   METRICS

We evaluate the learned DAGs using standard structure learning metrics, false discovery rate (FDR), true positive rate (TPR), false positive rate (FPR), and structural Hamming distance (SHD). In addition to these traditional metrics, we employ the Balanced Scoring Function (BSF), which adjusts the evaluation based on the relative difficulty of discovering edges or the absence of edges. BSF ranges from -1 (a completely incorrect graph) to 1 (a perfect match with the true graph).

## C   EDGE STRENGTH OF SINE FUNCTION: $L^2$ NORM FOR $f(x) = A\sin(Bx)$

Given $f(x) = A\sin(Bx)$, its derivative is $\frac{df}{dx} = AB\cos(Bx)$. The $L^2$ norm of $\frac{df}{dx}$ over a domain $\mathcal{X}$ is:

$$\left\| \frac{df}{dx} \right\|_{L^2(\mathcal{X})} = \left( \int_{\mathcal{X}} A^2 B^2 \cos^2(Bx)\, dx \right)^{1/2}.$$

Using the identity $\cos^2(Bx) = \frac{1+\cos(2Bx)}{2}$, the integral becomes:

$$\int_{\mathcal{X}} \cos^2(Bx)\, dx = \frac{1}{2}\mathrm{Vol}(\mathcal{X}),$$

where $\mathrm{Vol}(\mathcal{X})$ is the volume of the domain $\mathcal{X}$. Thus, the norm is:

$$\left\| \frac{df}{dx} \right\|_{L^2(\mathcal{X})} = AB\sqrt{\frac{1}{2}\mathrm{Vol}(\mathcal{X})}.$$

In linear models, the $L_2$-norm of the partial derivative is proportional to $\beta\sqrt{\text{Vol}(\mathbf{x})}$, where $\beta$ represents the edge strength. To ensure consistency with linear weights, we normalize the $L_2$-norm by the square root of the domain volume.

Thus, the final expression for the ground truth edge strength is:

$$W(f) := \frac{AB}{\sqrt{2}}.$$

## D    COMPARISON OF NEDAG-GP AND NOTEARS-MLP PERFORMANCE ACROSS DATA TYPES, GRAPH TYPES, AND NETWORK SIZES

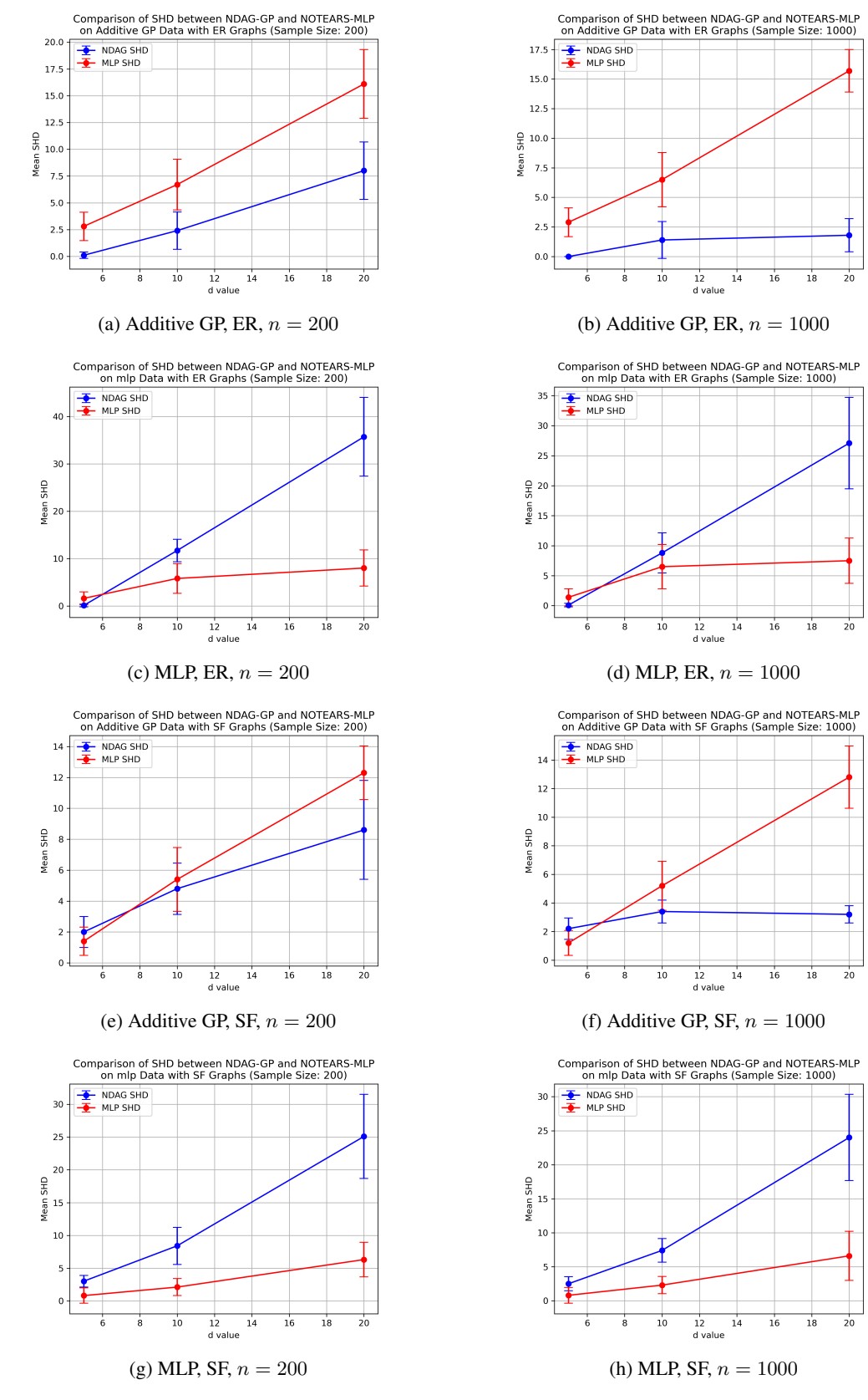

Figure 3: NEDAG-GP performs better on Additive GP data (blue), while NOTEARS-MLP excels on MLP data (red). As the network size increases (higher $d$-values), the task becomes more challenging, and both methods show higher SHD for larger networks. Results are shown for both $n = 200$ and $n = 1000$ sample sizes. Error bars represent std.

