# OpenReview forum: "Nonparametric Expert DAG Learning with Accurate Edge Strengths and Realistic Knowledge Incorporation"
_ICLR.cc/2025/Conference — Submitted to ICLR 2025_

### Official Review · Reviewer_sDxz · 2024-10-28

**Soundness:** 2
**Presentation:** 3
**Contribution:** 2
**Rating:** 3
**Confidence:** 4

**Summary:**

The paper studies directed acyclic graph (DAG) structure learning on observational data. It proposes NEDAG-GP, a new method that learns a nonparametric DAG as a Gaussian process (GP). NEDAG-GP also accommodates expert prior knowledge in the learned DAG.

**Strengths:**

DAG learning is undoubtedly an important problem for areas such as causal inference. The nonparametric nature of NEDAG-GP makes it appealing for complex nonlinear data, which is pervasive nowadays. Moreover, the capacity to incorporate expert knowledge is attractive. I also found the discussion around different characterizations of edge strength insightful.

**Weaknesses:**

1. The methodological innovation behind NEDAG-GP is limited. Specifically, the literature review indicates that GP-based DAG methods are already available, though NEDAG-GP sets up the weighted adjacency matrix differently. Moreover, incorporating expert knowledge is seemingly straightforward, though Section 4.2 does not actually explain how the knowledge-based constraints are enforced.
2. The paper's primary focus is unclear as it attempts to address two distinct problems simultaneously: nonparametric DAG learning and expert knowledge incorporation. Is there any reason expert knowledge cannot be included in linear DAGs, MLP-based DAGs, or spline-based DAGs? Or is there something particular about GP-based DAGs that makes them more amenable to integrating expert knowledge?
3. The experimental evidence in favor of NEDAG-GP (without expert knowledge) is limited. Figure 3 suggests that its good performance depends on whether the ground truth is a GP, so evaluations on a wider range of functions would be helpful. Also, DAGMA should be included as a baseline since it has superseded NOTEARS as the de facto DAG learning method in this area.
4. The paper does not provide a discussion or results about NEDAG-GP's uncertainty quantification performance, which is odd since it uses GPs.

**Questions:**

1. I found it strange that none of the in-text or reference list citations included years.
2. Equation 4: What is $H^1$? I could not see this set introduced anywhere.
3. Equation 5: Why is $x_k$ bold here? Other references to $x_k$ are not bold.
4. Section 4.1: It would help if $\sigma$ and $\ell$ are indexed by $j$ and $k$.
5. Section 5: This section is not substantive enough to constitute a single section. I suggest merging Section 5 with Section 6.
6. Table 2: How many replications are the results measured over?
7. Figure 2: There is no explicit reference to this figure anywhere in the text.
8. Appendix B.5: It would be helpful to provide the mathematical definitions of these metrics (or references to such). In particular, I am unfamiliar with the Balancing Scoring Function.
9. Figure 3: Each method is evaluated on a coarse grid of three points across the $x$-axis. It would be better to use a finer grid.

---

### Official Review · Reviewer_WgxK · 2024-10-29

**Soundness:** 3
**Presentation:** 3
**Contribution:** 3
**Rating:** 5
**Confidence:** 3

**Summary:**

This paper proposes a novel method for learning DAG structure based on continuous structure learning framework. Equipped with additive Gaussian Process with RBF kernel, this method provides non-parametric estimation of edge strengths and improving the interpretability of the structure learning process. The method also incorporates several types of expert knowledge, effectively enhances its performance.

**Strengths:**

This work is well-situated in the literature and fills the gap of utilizing non-parametric methods and incorporating expert knowledge in continuous structure learning framework. The advantages of the proposed method are supported by both synthetic experiment and real-world experiment.

**Weaknesses:**

1. Although I selected “good” for presentation, it would be better if the authors could include a pseudocode of their algorithm for a clearer presentation.

2. How to select the parameters of the Gaussian Processes? In supplementary B.1, the authors described the objective function, and it seems that the notation $\theta$ is unexplained. Does $\theta$ refer to the parameters of the Gaussian Processes? Also, It is still unclear to me how the expert knowledge is incorporated. Is it formulated as constraints of the optimization problem?

3. It seems that using non-parametric estimation method and incorporating expert knowledge make NEDAG-GP outperform NOTEARS-MLP. What if we compare NEDAG-GP with NOTEARS that is augmented with non-parametric estimation methods or expert knowledge incorporated? i.e. an ablation study.

**Questions:**

1. A recent paper [1] also discusses incorporating prior knowledge in continuous structure learning framework. Can the authors comment on the connections with paper [1]?


[1] Wang, Z,. Gao, X,. Liu, X,. Ru, X,. Zhang, Q,.(2024). Incorporating structural constraints into continuous optimization for causal discovery. Neurocomputing, Vol.595.

---

### Official Review · Reviewer_khLt · 2024-11-03

**Soundness:** 2
**Presentation:** 2
**Contribution:** 2
**Rating:** 3
**Confidence:** 3

**Summary:**

the paper proposes a GP process based continuous DAG learning framework. The approach is based on nonlinear DAG constraint from NOTEARS-MLP , utilizing the partial derivaties. Authors show prior knowledge can be incorporated into this framework. Empirical evaluation shows the proposed approach is better than NOTEARS-MLP.

**Strengths:**

the proposed approach to learn graph using GP partial derivative is new

improved performance over compared methods

**Weaknesses:**

- Unfortunately, the paper contain many imprecise statements (see below).

- only one method is compared, also ignoring many literatures on GP based causal models.

- motivation on using GP is not fully justified.

Other comments:
- " local decisions about adding, removing,or reversing edges are made without clear visibility into their global impact": this is not true, global consistency (and in some extend local consistency) properties of scores have been proven to show the optimality in these operations
- L65: is it true that a single number that can reveal the full caual relationships, esp they often come with specific distribution assumptions? In addition, score-based approach produce specific distribution scores, constraint-based approaches offer test stats, which all represent edge weights.
- L144: The knowledge on edge weights can be easily be via regularization, such as the L1 sparsity coefficient to achieve confidence in forbidden edges. The objective itself is data fitting + prior as regularizations. Topological order itself can be expressed by a set of forbidden edges.
- Section 4.2: I don't see how these W constraints can not be expressed by existing continuos learning approaches. In addition, exppressing prior knowledge as an exact numerical value seems harder

**Questions:**

- it has been known GP and NNs share at least similarities, for example "DEEP NEURAL NETWORKS AS GAUSSIAN PROCESSES" ICLR 2018. However, the proposed approach did not fully explore and differentiate the use of GP from NNs, besides just a nonparametric approach in name. It would be good that authors can show, in some theoretical statement, where GP based dag learning can be superior.
- Some related works on causal graphs and gaussian process are not discussed and compared. e.g.,
Aglietti et al, "Multi-task Causal Learning with Gaussian Processes".
Wilson et al, "Gaussian Process Regression Networks".
- typical distribution assumptions are needed to guarantee identifiablity. What can be guaraneted, in term of the identifiability or consistency, for the proposed method?

---

### Official Review · Reviewer_gHXx · 2024-11-04

**Soundness:** 2
**Presentation:** 2
**Contribution:** 1
**Rating:** 1
**Confidence:** 4

**Summary:**

This paper proposes a nonparametric method for quantifying edge strength and incorporating domain knowledge into modeling. It builds upon the well-known NOTEARS causal discovery method, which transforms the combinatorial search process into a continuous optimization problem. By leveraging nonparametric techniques such as Gaussian Processes, the NEDAG-GP method offers interpretable weights within a nonparametric modeling framework.

**Strengths:**

The author provides a highly intuitive introduction to the background and existing challenges in the field, making it accessible even for readers less familiar with the topic. The writing style is clear and straightforward, which enhances comprehension. The explanations are both precise and easy to follow, contributing to a well-structured presentation of ideas. The paper presents both qualitative and quantitative experimental results that are insightful and visually intuitive, aiding in understanding the effectiveness of the proposed method. Additionally, the inclusion of the Sachs dataset as a real-world example is particularly informative, demonstrating the practical applicability of the method and adding significant value to the study.

**Weaknesses:**

(1) While the paper aims to address DAG learning for modeling causal structures and complex dependencies, the purpose could be clarified. It seems that causal structures inherently involve complex dependencies, so it would help to clarify how these terms are being distinguished in the context of this work. If the intent is to use a DAG for causal reasoning, some discussion on the identifiability of the learned DAG would strengthen the contribution. Specifically, it would be helpful to know if the learned DAG represents a unique solution given the data or if it belongs to an equivalence class that includes the ground truth. Reviewing classic works on causal discovery algorithms, such as PC, GES, or PNL, could help refine the objectives and theoretical foundation of the approach.

(2) The paper introduces the idea of incorporating Gaussian Processes into DAG learning, leveraging their nonparametric properties. While this is an interesting direction, the novelty may be somewhat limited, as Gaussian Processes are a known approach for handling nonparametric modeling. Given an adjacency matrix with binary indicators, there are many established methods for estimating associated parameters, so it would be valuable to see a discussion on how this approach contributes uniquely to the field.

(3) Some aspects of the writing could be more clear. For instance, in the introduction, two bolded statements emphasize the importance of incorporating expert knowledge while minimizing reliance on expert-specified parameters and distributional assumptions. Since expert knowledge can encompass information on edges, parameters, and distributions, it would help to clarify the intended balance between these elements. Addressing this and similar points throughout the paper would enhance readability and help readers better understand the author’s perspective and familiarity with the field.

**Questions:**

(1) Could you elaborate on the purpose of this paper? For instance, why is DAG learning needed, and how does it differ from causal discovery?

(2) Could you provide more evidence of the significance of your work beyond its ability to incorporate expert knowledge and quantify edge strengths?

(3) Could you explain why you are confident in the accuracy of the learned edge strengths? What makes them reliable, and how would you convince others to use them in downstream tasks?

(4) Could you clarify why classic causal discovery algorithms are not mentioned or compared in your paper? Additionally, why is continuous learning preferable to traditional score-based, combinatorial structure learning methods? I am not fully convinced by your statement that “In combinatorial search, local decisions about adding, removing, or reversing edges are made without clear visibility into their global impact, only revealed once the global objective is minimized,” as this issue is specifically addressed by GES.

---

### Meta-Review · Area_Chair_NEG5 · 2024-12-21

**Metareview:**

The authors introduce NEDAG, a novel method for quantifying edge strength and incorporating domain knowledge into the continuous structure learning framework. All reviewers vote to reject and there is no response from the authors.

**Additional Comments On Reviewer Discussion:**

No author response.

---

### Decision · Program_Chairs · 2025-01-22

Reject